# Squalene Epoxidase Correlates E-Cadherin Expression and Overall Survival in Colorectal Cancer Patients: The Impact on Prognosis and Correlation to Clinicopathologic Features

**DOI:** 10.3390/jcm8050632

**Published:** 2019-05-08

**Authors:** Joo Heon Kim, Chang Nam Kim, Dong Wook Kang

**Affiliations:** 1Department of Pathology, Eulji University Hospital, Eulji University School of Medicine, Daejeon 35233, Korea; kjh2000@eulji.ac.kr; 2Department of Surgery, Eulji University Hospital, Eulji University School of Medicine, Daejeon 35233, Korea; kimcn@eulji.ac.kr

**Keywords:** squalene epoxidase (SE), *SQLE*, E-cadherin, immunohistochemistry, colorectal cancer

## Abstract

Squalene epoxidase (SE), coded by *SQLE,* is an important rate-limiting enzyme in the cholesterol biosynthetic pathway. Recently, the aberrant expression of *SQLE*, which is responsible for epithelial to mesenchymal transition (EMT), has been reported in various types of cancer. This study was undertaken to clarify the clinicopathologic implications of SE in patients with stage I to IV colorectal cancer (CRC). We also analyzed the expression patterns of SE in association with E-cadherin in a series of CRCs. We detected the cytoplasmic expression of SE in 59.4% of carcinoma samples by immunohistochemistry (IHC). There was a significant correlation between a high level of SE expression and lymphovascular (LV) invasion (*p* < 0.001), tumor budding (*p* < 0.001), invasion depth (*p* = 0.002), regional lymph node metastasis (*p* < 0.001), and pathologic TNM stage (*p* < 0.001). SE is more abundantly expressed at the invasive front, and reversely correlated with E-cadherin expression. Patients with SE-positive CRC had shorter recurrence-free survival (RFS) and poor overall survival (OS) than those with SE-negative CRC in multivariate analysis (*p* < 0.001 and *p* < 0.001, respectively). These data suggest that SE can serve as a valuable biomarker for unfavorable prognosis, and as a possible therapeutic target in CRCs.

## 1. Introduction

Colorectal cancer (CRC) ranks fourth highest in the number of all cancer cases reported worldwide and is currently one of the leading causes of cancer-associated mortality [1,2], despite earlier detection and many accomplishments in the development of molecular-based treatment. CRC is a heterogeneous and complex disease, and its molecular mechanism of carcinogenesis is a multi-step process related to the genomic instability associated with genetic alterations [3]. Furthermore, knowledge of the invasion and metastatic process of colon cancer is still very limited. Thus, the discovery of tumor markers is highly desirable, as it would allow treatment to be tailored to individuals likely to require better-targeted therapies in clinical practice. The human squalene epoxidase (SE), coded by *SQLE* which maps to chromosomal 8q24.13, catalyzes the conversion of squalene to 2,3(S)-oxidosqualene in cholesterol biosynthesis and is suggested to be one of the critical rate-limiting enzymes in downstream cholesterol biosynthesis [4,5]. It has been exploited as a target for the development of hypocholesterolemic and antifungal agents. Since 1981, several studies have demonstrated that *SQLE* is deregulated in human cancers and has multifunctional roles. Studies conducted using cDNA microarrays reported the presence of *SQLE* in sets of differentially expressed transcripts in breast cancers [6] and lung squamous cell carcinoma [7]. Recent studies have shown that *SQLE* is differentially expressed in lung cancer [7,8], breast cancer [9,10], prostate cancer [11,12], hepatocellular carcinoma [13], and esophageal cancer [14]. *SQLE* overexpression or copy number variation is significantly associated with tumor progression, metastases and unfavorable outcomes in prostate cancer and breast cancer patients [6,15,16]. In vitro analysis showed that *SQLE* knockdown reduced the migration, progression, invasion, and metastasis of prostate cancer cells [11], and diminished epithelial to mesenchymal transition (EMT) in esophageal cancer [17]. Higher *SQLE* expression was strongly associated with increased histologic markers of angiogenesis [18]. Even though *SQLE* may still influence tumorigenesis and tumor progression, its intracellular function and regulation in tumor cells remain to be further investigated. To date, there is no report regarding the prognostic value of abnormal SQLE protein expression in colorectal cancer tissues. Therefore, we explored the clinicopathologic correlation and its clinical significance in CRC and correlation between SE and E-cadherin expression status in CRC. 

## 2. Materials and Methods 

### 2.1. Case Selection

A series of 143 cases of CRC were selected from patients who underwent surgical resection at Eulji University Hospital from 2000 to 2005. We excluded those specimens obtained from patients who received preoperative neoadjuvant chemoradiotherapy. The pertinent clinical and pathologic information was obtained from pathology reports and electronic operation records. All cases were histologically confirmed as primary colorectal adenocarcinoma, and the H&E slides were re-examined by two independent pathologists (Kim, J.H. and Kang, D.W.). The tumor grade was classified into a low grade (≥50% of tubules) and high grade (<50% of tubules) [19]. Tumor budding was defined as a single tumor cell or a group of up to four detached cancer cells, and classified into two grades [20,21,22]. The cancer recurrence was indicated as a tumor presenting at the anastomosing site, in the perineum or pelvic cavity and regional lymph nodes diagnosed by radiologic finding, colonoscopy and exploratory surgical and/or microscopic examination. Also, tumor metastasis was designated as the presence of cancer cells outside the area of surgical resection, including the lung, liver, pancreas, bone and other organs. 

### 2.2. Immunohistochemistry (IHC)

For the IHC, all cases of CRC tissue with accompanying normal epithelium were fixed in 10% buffered formalin for 24 to 48 h and embedded in paraffin. Immunostaining was performed on tissue sections of 4 µm thickness, using IgG-rabbit polyclonal antibody against SQLE (1:100 dilution; Interchim, Montluçon, France) and mouse monoclonal antibody against E-cadherin (NCH-38, 1:100 dilution; DakoCytomation, Glostrup, Denmark) as primary antibodies. Paraffin-embedded tissue sections were deparaffinized and rehydrated through a series of xylene and graded ethanol, and autoclaved at 120 °C for 10 min with 10 mM/L sodium citrate buffer (pH 6.0). IHC conditions for SE and E-cadherin were optimized according to the manufacturers’ instructions. The slide sections were incubated with the primary antibodies for 90 min and stained with 3,3′-diaminobenzidine as the substrate using an EnVision-HRP kit (DakoCytomation, Glostrup, Denmark). Negative controls were obtained by using an irrelevant mouse IgG of the same isotype. All slides were counterstained for 1 min using Mayer’s hematoxylin and then mounted.

### 2.3. Assessment of Immunohistochemical Staining 

To evaluate the immunohistochemical expression of SE and E-cadherin in association with the various clinicopathologic parameters, the immunoreactivity of both SE and E-cadherin were analyzed in a semi-quantitative manner by two independent pathologists, who were blinded to outcome. Immunoreactivity for SE and E-cadherin was observed primarily in the cytoplasm and cytoplasmic membrane of the normal colonic epithelium and tumor cells, respectively. The intensity of immunohistochemical stain was scored as 0, 1 and 2 (0: no or less weak staining than normal colonic epithelium, 1: stained similar to the normal colonic epithelium, and 2: stronger staining than the normal colonic epithelium). The proportion of positive tumor cells was scaled as 1 to 4 (1: 0–24% of tumor cells, 2: 25–49% of tumor cells, 3: 50–74% of tumor cells and 4: 75–100% of tumor cells, respectively) [23]. To evaluate the statistical analysis between SE and E-cadherin expression and the clinicopathologic parameters, the cutoff value (25% of tumor cells showing a strong immunoreactivity than normal colonic epithelium) was used to distinguish between the low expression (<25% of tumor cells) and high expression (>25% of tumor cells). Conflicting cases were re-examined, and consensuses were reached.

### 2.4. Statistical Analysis of Prognostic Parameters 

We performed statistical analyses using the SPSS software package (ver. 21; SPSS Inc., Chicago, IL, USA). The correlation between SE and the various clinicopathologic parameters was analyzed with Pearson’s chi-square test or Fisher’s exact test and one-way ANOVA test. To evaluate the statistical significance, recurrence-free survival (RFS) was defined as the time from the date of surgical operation to the first date of recurrence, or the date of the last follow-up. Similarly, overall survival (OS) was defined as the duration from the date of surgery to the date of death, or the date of the last follow-up. The mean follow-up duration for all of the patients was 65.0 months, ranging from 0.6 to 184.9 months. Using the Kaplan–Meier (product-limit) test, the RFS curve, and the OS curve were formulated. Multivariate analysis for OS and RFS was also performed with Cox proportional hazard regression analysis. To examine the statistical correlation between the differences in survival distribution, the log-rank test was used. In all statistical analyses, *p*-values less than 0.05 were considered statistically significant. 

### 2.5. Ethical Permission

The Institutional Review Board of Eulji University Hospital approved the study protocol and provided all necessary ethical permission (IRB File No. 2019-02-014-001). 

## 3. Results

### 3.1. Association of SE and E-Cadherin Expression Status with Clinicopathologic Characteristics 

Expression levels of SE and E-cadherin were evaluated by immunohistochemical analysis. The high expression of SE and E-cadherin expression at the invasive front was observed in 85 (59.4%) and 17 (11.9%) of the 143 patients, respectively. SE immunoreactivity was found primarily in the cytoplasm of the cancer cells. E-cadherin staining was seen predominantly in the cytoplasmic membrane, as previous studies have shown. In normal colonic epithelium, immunoreactivity for SE was mostly none or weak. Expression levels of SE were highest in the tumor invasive front cluster, lower in the tumor center, and lowest in normal epithelium. In contrast, expression levels of E-cadherin increased significantly in normal epithelium and decreased from tumor center to tumor invasion front clusters. Figure 1 shows representative expression patterns of SE and E-cadherin in CRC. The clinical and pathologic characteristics of the 143 CRC patients who underwent surgical resection are summarized in Table 1. The median age of the CRC patients at the time of surgical operation was 62.2 years (M:F = 75:68, ranging: 28–86 years) and the median tumor size was 5.2 cm (range: 0.8–12.0 cm) in maximum tumor diameter. We analyzed whether the SE expression level was associated with clinicopathologic factors potentially predictive of prognosis. Patients with high SE expression at the invasive front showed a significantly greater presence of lymphovascular (LV) invasion, deeper invasion depth (pT stage), more frequent regional lymph node metastasis (pN stage), and more advanced tumor staging (pTNM stage) than in those with low SE expression levels. We also evaluated the association between E-cadherin expression levels and these variables. A statistically reverse correlation was also found between E-cadherin expression status and SE expression level (*p* < 0.001) (Figure 2). Specifically, of the 85 cases exhibiting a high level of SE expression at the invasive front, a total of 71 cases (83.5%) showed decreased membranous immunoreactivity of E-cadherin. Of the 58 cases with a low level of SE expression, only 3 cases (5.2%) revealed increased E-cadherin expression. In addition, of the 60 cases (42.0%) with high expression of SE in the tumor center, 47 cases (73.8%) revealed the loss of immunoreactivity of E-cadherin. The immunohistochemical comparison between SE and E-cadherin expression in the invasive front and tumor center is summarized in Table 2.

### 3.2. High SE Expression at the Invasive Front Correlates with RFS and OS 

We performed a univariate analysis of whether the expression status of SE correlates with RFS and OS. Forty-one (28.6%) patients presented with cancer recurrence during follow-up and 62 (43.3%) patients died of CRC with or without metastasis. Seven (4.9%) patients died of unknown causes, 11 (7.6%) patients were alive with local recurrence and/or distant metastasis, and 63 (44.0%) patients remained alive and recurrence-free. The Kaplan–Meier analysis showed that there was a significant correlation between high SE expression and reduced RFS (*p* < 0.001) (Figure 3A). RFS was shorter in patients with high expression levels of SE, being only a mean duration of 53.60 months (95% confidence interval (CI), 39.298–67.909), whereas it was longer in those patients with low levels of SE expression, being a mean duration of 143.36 months (95% CI, 124.453–162.270). There was also a significant correlation between high SE expression and shorter OS (*p* < 0.001). SE expression status significantly split the cumulative OS curves of patients (Figure 3B). While the median OS for CRC patients with a high level of SE expression was 70.8 months (95% CI, 55.603–85.921), the median OS of CRC patients with a low SE expression level was increased to 157.0 months (95% CI, 141.161–172.760). A multivariate analysis was also done to assess the predictive value of SE expression for RFS and OS by adjusting other potentially prognostic parameters. In a multivariate Cox regression analysis, SE expression status was an independent prognostic factor significantly associated with RFS with a *p*-value of <0.001. The relative risk (RR) of tumor recurrence for patients with a high SE expression level was 3.647 (95% CI: 1.912–6.955). A high level of SE expression was also predictive of reduced OS (*p* < 0.001). The RR of death in patients with a high level of SE expression was more than three times greater (RR: 3.976; 95% CI: 1.894–8.347) than in those with low SE expression levels. In addition to SE expression status, statistically significant clinicopathologic factors that were correlated with OS were invasion depth (*p* = 0.030), and distant metastasis (*p* < 0.001). Table 3 summarizes the results from the Cox proportional hazards analysis. 

## 4. Discussion 

SE belongs to the flavoprotein monooxygenase family, which catalyzes a wide variety of oxidative reactions and epoxidation [24,25] in cholesterol biosynthesis, and is differentially expressed in various human solid cancers. Alterations in metabolism are critical for the invasion and metastatic process of the tumor. However, the mechanisms by which these metabolic changes are controlled by the major drivers of the tumorigenic process remain elusive. Haider et al. confirmed SE as a metabolic driver in multiple cancers, demonstrating its association with poor prognosis and tumor hypoxia [26]. Thus, the *SQLE* gene may act as an oncogene, but whether it plays a role in colorectal cancer remains unknown. Herein, our study shows that SE expression was higher in the CRC tissues than in normal tissues. There is a significant correlation between the overexpression of SE and invasion depth, lymph node metastasis, and pTNM stage. 

Cholesterol is an essential biological component required for the structural integrity of the cell membrane and lipid rafts and is also a precursor to steroid hormones. Cancer cells have an increased demand for cholesterol synthesis and metabolism, which is needed in tumor growth and progression. SE is an important cholesterol biosynthetic enzyme for the regulation of cellular cholesterol homeostasis in the endoplasmic reticulum. This enzyme is controlled by different mechanisms. One is through direct synthesis via the transcription of sterol regulatory element-binding proteins, which are activated in response to low sterol status, and bind sterol regulatory element consensus sequences [27]. In addition to transcriptional regulation, the rapid shutdown of cholesterol synthesis requires post-transcriptional control. *SQLE* is directly regulated by cholesterol itself [28,29]. Although the pathologic role of cholesterol in carcinogenesis is not fully understood and lacks consensus, recent reports show that intracellular de novo cholesterol biosynthesis is significantly upregulated in cancer cells, or the enzyme activity that catalyzes the rate-limiting step in de novo cholesterol biosynthesis is augmented in cancer cells, especially under hypoxia [30,31,32,33]. Cholesterol can be enzymatically modified to form metabolites, such as a family of oxysterols [cholesterol oxidation products (COP)]. COPs modulate the activity of signal transduction cascades [34,35,36]. For instance, 27 hydroxycholesterol (27-OHC) is a selective estrogen receptor modulator and an agonist of the liver X receptor. It is involved in tumor cell proliferation, EMT, tumor cell invasion, migration, and metastasis in breast cancer and prostatic cancer [37,38]. Therefore, it is necessary to study how *SQLE* contributes to the structural modification of the plasma membrane, and the formation of various COPs during tumorigenesis of CRC. 

In the present study, we observed that there was a significant correlation between a high expression level of SE and the tumor budding status and invasion depth in the CRC. Tumor budding has been suggested to be associated with EMT, as evidenced by decreased or aberrant expression of E-cadherin [39,40]. The SE and E-cadherin protein expression pattern in the tumor budding of the CRC was determined by immunohistochemical analysis. SE expression significantly increased in the invasive front, especially in the tumor budding of the invasive front in contrast to the tumor center of the CRC. However, E-cadherin expression decreased in the corresponding infiltrating tumor budding cells. There was a reverse correlation between the expression level of SE and E-cadherin expression levels. These results imply that *SQLE* might be involved in the modulation of EMT, resulting in tumor invasion and progression. Although further studies are necessary to validate our findings, we suggest that *SQLE* may be an EMT-associated marker of the CRC. EMT is a reversible genetic program of trans-differentiation of epithelial cells into mesenchymal cells, and cancer cells acquire EMT/MET plasticity in the process of carcinogenesis and tumor progression. This phenomenon is affected by tumor cell metabolism and tumor micro-environmental factors including hypoxia and growth factors, and it permits tumor cells to involve the reorganization of the signaling network that governs tumor cell survival, proliferation, and homeostasis. Consistent with our results, Qin Y et al. reported that *SQLE* induces EMT by regulation of miR-133b in esophageal squamous cell carcinoma [17]. The loss of E-cadherin and the acquisition of a more mesenchymal phenotype have been shown to correlate with clinically poor prognosis and metastasis in various epithelial-derived solid tumors. Tumors with EMT gene signatures were more resistant to anti-cancer drugs and a significantly longer time to progress in patients enrolled in a randomized non-small cell lung cancer clinical trial [41]. Herein, our study shows that there is a significant correlation between a high expression level of SE and tumor budding, and a decrease in E-cadherin in the invasive front. Moreover, there is a significant correlation between high SE expression and tumor poor RFS, and worse OS in CRC patients. In CRC, tumor metastasis is the most frequent cause of treatment failure and is responsible for 90% of patient mortality. However, there is no molecular marker that can adequately predict the risk of tumor progression and metastasis. Taken together, the results suggest that SQLE may be considered a valuable biomarker for unfavorable prognosis, and could be the basis of a new strategy to target cholesterol metabolism for treating CRC.

## 5. Conclusions 

The study aimed to obtain confirmative information for the clinicopathological significance and prognostic role of SE expression in human colorectal cancer. There was found to be a significant correlation between a high level of SE expression and LV invasion, tumor budding, invasion depth, lymph node metastasis, and pathologic TNM stage. In addition, a reverse correlation was found between SE and E-cadherin expression in the invasive front in the colorectal cancer tissue. High expression of SQLE can serve as a valuable biomarker for unfavorable prognosis and as a possible therapeutic target in CRCs

## Figures and Tables

**Figure 1 jcm-08-00632-f001:**
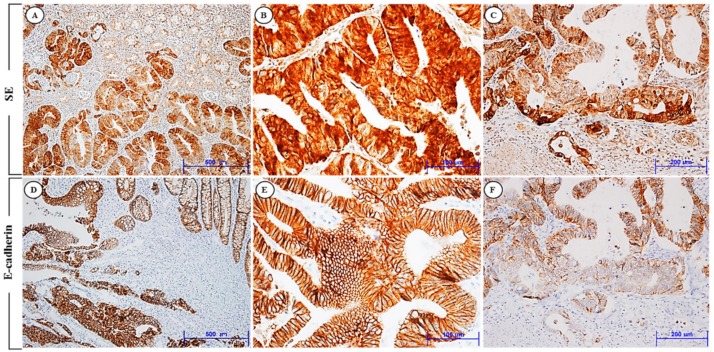
Immunohistochemical expression of SE (**A**–**C**) and E-cadherin (**D**–**F**) in human CRC. (**A**) Tumor cells show high SE expression, but no or weak expression of SE in the normal colonic epithelium (×100). (**B**) Tumor cells reveal strong SE expression primarily in the cytoplasm of the tumor cells (×400). (**C**) SE expression is highly increased in the tumor cells of the invasive front (×200). (**D**) Strong immunoreactivity of E-cadherin in the normal colonic epithelium and tumor cells (×100). (**E**) E-cadherin is highly expressed, predominantly in the membrane of the tumor cells (×400). (**F**) E-cadherin expression decreases from tumor center to tumor invasion front clusters (×200). SE and E-cadherin show different immunohistochemical expression in the invasion front of human CRC.

**Figure 2 jcm-08-00632-f002:**
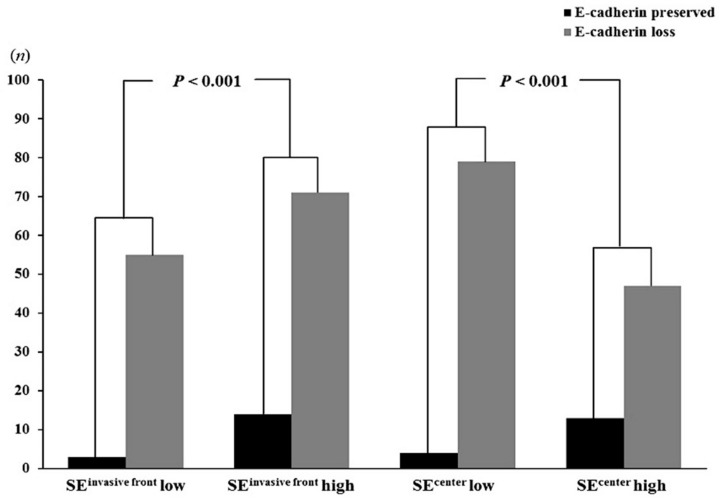
Immunohistochemical relationship between SE and E-cadherin in CRC. The expression pattern of SE is reversely correlated with E-cadherin expression in both the invasive front and the tumor center of CRC (*p* < 0.001).

**Figure 3 jcm-08-00632-f003:**
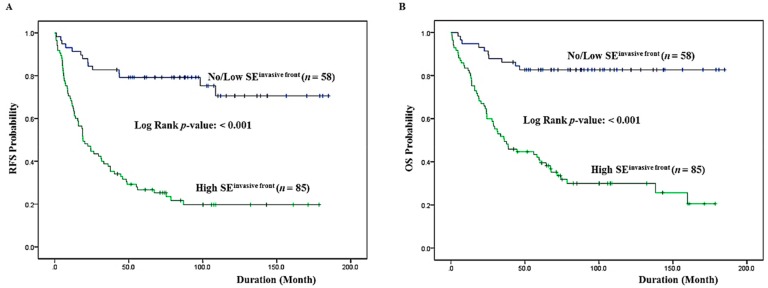
Kaplan–Meier survival analysis by SE expression status at the invasive front. (**A**) Cumulative RFS differences between patients with high and low SE expression. (**B**) Cumulative OS differences between patients with high and low SE expression. The *p*-value was obtained using the log-rank test of the differences. RFS: recurrence-free surviva; OS: overall survival.

**Table 1 jcm-08-00632-t001:** Clinicopathologic variables and the expression status of SE at the invasive front in CRC.

Characteristics	Total	SE Expression				*p*
		Negative/Low		High		
		*n* = 58	%	*n* = 85	%	
Age (years)						
<50	26	10	17.2	16	18.8	0.830 *
≥50	117	48	82.8	69	81.2	
Gender						
Female	68	33	48.5	35	33.3	0.088 *
Male	75	25	51.5	50	66.7	
Site						
Right/Transverse colon	34	16	27.6	18	21.2	0.426 *
Left colon and rectum	109	42	72.4	67	78.8	
Size						0.391 *
<5 cm in diameter	60	27	46.6	33	38.8	
≥5 cm in diameter	83	31	53.4	52	61.2	
Grade						1.000 *
Low	111	45	77.6	66	77.6	
High	32	13	22.4	19	22.4	
LV invasion						**<0.001 ***
Not identified	39	26	44.8	13	15.3	
Present	104	32	55.2	72	64.7	
Tumor border						0.378 *
Pushing	13	7	12.1	6	7.1	
Infiltrating	130	51	87.9	79	92.9	
Tumor budding						**<0.001 ***
Low	36	26	44.8	10	11.8	
High	107	32	55.2	75	88.2	
Invasion depth						**0.002 ^+^**
pT1	5	4	6.9	1	1.2	
pT2	21	15	25.9	6	7.1	
pT3	105	36	62.1	69	81.2	
pT4	12	3	5.2	9	10.6	
LN metastasis						**<0.001 ^‡^**
pN0	67	37	63.8	30	35.3	
pN1	23	11	19.0	12	14.1	
pN2	53	10	17.2	43	50.6	
Distant metastasis						**0.005 ^+^**
M0	121	55	94.8	66	77.6	
M1	22	3	5.2	19	22.4	
TNM stage						**<0.001 ^‡^**
I	21	16	27.6	5	5.9	
II	45	21	36.2	24	28.2	
III	55	18	31.0	37	43.5	
IV	22	3	5.2	19	22.4	

SE, squalene epoxidase; LV, lymphovascular invasion; LN, lymph node. * *p* values were estimated by Pearson’s chi-square test; ^+^
*p* values were estimated by Fisher’s exact test; ^‡^
*p* values were estimated by one-way ANOVA test; *p* < 0.05 are highlighted in bold.

**Table 2 jcm-08-00632-t002:** Comparison of expression between SE and E-cadherin in CRC.

SE Expression	E-Cadherin Expression		*p*-Value
	High (*n* = 17) (%)	Low/Negative (*n* = 126) (%)	
Invasive front			
Low/negative (*n* = 58)	3 (5.2)	55 (94.8)	**<0.001**
High (*n* = 85)	14 (16.5)	71 (83.5)	
Tumor center			
Low/negative, (*n* = 83)	4 (4.8)	79 (95.2)	**<0.001**
High, (*n* = 60)	13 (21.7)	47 (73.8)	

SE, squalene epoxidase. *p* values were estimated by Fisher’s exact test and *p* < 0.05 are highlighted in bold.

**Table 3 jcm-08-00632-t003:** Multivariate analysis for RFS and OS in CRC.

	*n*	RFS		OS	
		Relative Risk (95% CI)	*p*	Relative Risk (95% CI)	*p*
SE^invasive front^			**<0.001**		**<0.001**
Low/negative	58	1.000		1.000	
High	85	3.647 (1.912–6.955)		3.976 (1.894–8.347)	
LV invasion			0.768		0.939
Not identified	39	1.000		1.000	
Present	104	0.911 (0.490–1.694)		0.973 (0.490–1.934)	
Budding			0.657		0.882
Low	36	1.000		1.000	
High	107	1.153 (0.614–2.165)		0.949 (0.476–1.893)	
Invasion depth			**0.022**		**0.030**
pT1 + pT2	26	1.000		1.000	
pT3 + pT4	117	3.774 (1.314–10.840)		3.872 (1.138–13.175)	
LN metastasis			0.833		0.223
Not identified	67	1.000		1.000	
Present	76	1.252 (0.747–2.099)		1.433 (0.804–2.554)	
Distant metastasis			**0.001**		**<0.001**
M0	121	1.000		1.000	
M1	22	2.592 (1.455–4.618)		3.569 (1.935–6.523)	

RFS, recurrence-free survival; OS, overall survival; LV, lymphovascular invasion; LN, lymph node; CI, confidence interval. *p* values were obtained by Cox proportional hazards analysis and *p* < 0.05 are highlighted in bold.

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
