# Peer review of "Squalene Epoxidase Correlates E-Cadherin Expression and Overall Survival in Colorectal Cancer Patients: The Impact on Prognosis and Correlation to Clinicopathologic Features"

_jcm, 2019, doi:10.3390/jcm8050632_

Reviewer 1 Report

Summary

In the manuscript entitled “Squalene epoxidase inversely correlates E-cadherin expression and prolongs overall survival in colorectal cancer patients: The impact on prognosis and correlation to clinicopathologic features” the authors aimed at investigating the potential association of squalene epoxidase (SQLE) and E-cadherin expression with clinical features and prognosis of colon cancer. They showed that SQLE is highly expressed in a significant number of biopsies of colon cancer patients whose OS is reduced and cancer features (e.g. invasion, tumor budding) are more severe. Interestingly, E-cadherin expression is inversely correlated with the levels of SQLE. Overall, the authors conclusion is that SQLE is remarkably associated with unfavourable prognosis of colon cancer. Consequently, they propose SQLE as potential prognosis marker.

Overall, the authors conducted a well-designed study which can add new insight in the relevant field of study, improving the knowledge about genes potentially involved in the pathogenesis of colon cancer. Also, there are no so many studies about the over-mentioned topic. The impact of this manuscript is potentially good.

General comments

This manuscript is well written and the scientific question to be addressed is clearly stated; the abstract and introduction are generally well organized and include the relevant references to literature. Particularly, the authors chose a science sound experimental approach where the conclusions are well supported by the results.

However, there are some minor concerns about the presentation of the results data, and the description of the tables and figures can further be improved. Overall, the language is good but some improvements can be done.

See specific comments below.

Specific comments

Line 34-37: this sentence is very long and twisted. The authors should re-write it in a more concise and clear way for a better reading-flow;

Line 42-43: the start of the sentence “SQLE overexpression in gene expression or copy number…” can be improved.

Table 1-3: the legends of the tables could be improved adding at least a longer methodological description (in addition to the one already present in the methods). For instance, they could improve the explanation/description of statistical analysis and percentage;

Figure 1: The name of the analysed protein (SQLE or E-cadherin) could be added by the first picture of each raw (A-C and D-F);

Line 111: “The highly elevated expression of…” -> you can use “the high expression of…”.

Author Response

Response to Reviewer 1 Comments

Point 1: Line 34-37: this sentence is very long and twisted. The authors should re-write it in a more concise and clear way for a better reading-flow;

Response 1: As the reviewer recommended, we re-wrote the sentence.

“Thus, the discovery of tumor markers is highly desirable, as it would allow treatment to be tailored to individuals likely to require better-targeted therapies in clinical practice.”

Point 2: Line 42-43: the start of the sentence “SQLE overexpression in gene expression or copy number…” can be improved.

Response 2: As the reviewer has mentioned, we changed the sentence in red color.

Point 3: Table 1-3: the legends of the tables could be improved adding at least a longer methodological description (in addition to the one already present in the methods). For instance, they could improve the explanation/description of statistical analysis and percentage;

Response 3: As the reviewer pointed out, we added a note and methodological description of the table.

Point 4: Figure 1: The name of the analysed protein (SQLE or E-cadherin) could be added by the first picture of each raw (A-C and D-F);

Response 4: As the reviewer pointed out, we added labels of squalene epoxidase (SE) and E-cadherin, scale bars and alphabets.

Point 5: Line 111: “The highly elevated expression of…” -> you can use “the high expression of…”.

Response 5: As the reviewer pointed out, we changed the phrase in red color

We tried to address the points raised by the reviewers as best as we can. The specific responses to the reviewers’ comments are described below. We also fixed other unintended errors on our manuscript.

Once again, we are thankful to have a chance to improve our manuscript and hope that it is good enough to be published in the Journal of Clinical Medicine

Please let us know if there are any more things we need to change.

Thank you again.

Reviewer 2 Report

Colorectal cancer (CRC) displays a major health burden and metastasis and disease recurrence remain challenging. Epithelial-to-mesenchymal transition (EMT) was described as one factor contributing to metastasis. In their study, the authors aimed to elucidate the correlation between the EMT key player E-cadherin and the enzyme Squalene Epoxidase (SQLE) in CRC patients.  For this purpose, they have used a comprehensive approach by evaluating E-cadherin and SQLE immunohistochemical staining on human CRC biopsies and correlated the staining intensities to survival and further clinical features. Their findings on this topic indicate the relevance of SQLE in CRC and will be of interest to the readership of the Journal of Clinical Medicine. A number of major and minor points should be addressed prior to publication as detailed below.

Was the specificity of the antibodies used for IHC tested? For instance, did one specific band occur when performing western blot? This question came up when TCGA survival data of CRC patients were checked on the Human Protein Atlas website. Here, it seems like low SQLE expression is associated with poor prognosis. How do the authors explain this discrepancy?

In general, many results were provided in the text (percentages, n-numbers) or in tables (e.g. Table 2, Table 3, section 3.2) but not in figures. However, it would be helpful to the readers if these results would also be visualized. For instance, the finding outcome of the assessment of the IHC could be shown in pie charts displaying: the IHC intensity (score 0-2) and the proportion of positively stained tumor cells (score 1-4) as described in section 2.2 (line 83-87). Another example would be Table 3 where the results could be visualized as a Venn diagram.

Can the authors provide survival data correlated to total SQLE staining intensity? In Figure 2 only survival data associated with high/low SQLE staining at the invasive front were shown. How was the survival affected when the staining in the complete tumor section (and not only the invasive front) was considered?

In Table 1 clinical features were associated to negative/low and high SQLE levels. Do these levels refer to the staining intensity at the invasive site of the tumor of at the total staining intensity?

It is mentioned that “Figure 1 and 2 show representative expression patterns of SQLE and E-cadherin in CRC.” (line 118-119), however, Figure 2 displays survival data.

How long have the sections been autoclaved at 120°C when performing IHC?

Please provide the registration number of the ethical permission.

The upper half of the Discussion (line 181-212) contains a lot of theoretical background on SQLE which (at least in some parts) should be moved to the Introduction section.

Please add n-numbers to the Kaplan-Meier curves.

The number of CRC patients analyzed in this study (n=143) should be mentioned in the Materials and Methods section.

There is no section “2.1” in the Materials and Methods section. The first part of the Materials and Methods section could be subdivided into “2.1 Sample acquisition and tumor grading” and “2.2 Immunohistochemistry” which are then followed by “2.3 Assessment of immunohistochemical staining”.

Add labels to Figure 1: On the left side of A) the label “SQLE” and on the left side of D) “E-cadherin” could be added. In addition, scale bars are missing.

Please revise the Acknowledgments section “Acknowledgments: In this section you can acknowledge any support given which is not covered by the author contribution or funding sections. This may include administrative and technical support, or donations in kind (e.g., materials used for experiments).” (line 252-254).

Please define meanings of acronyms the first time they are being used.

The authors did not pay attention to the nomenclature. Human gene names should be written in capital letters and be italicized. Human proteins are written in capital letters. Murine gene names should be written in small letters (only the first letter is capitalized) and be italicized. Murine proteins are written in capital letters.

Author Response

We thank the reviewers for their constructive comments on our manuscript  (Manuscript ID: jcm-500556)

We tried to address the points raised by the reviewers as best as we can. The specific responses to the reviewers’ comments are described below. We also fixed other unintended errors on our manuscript.

Once again, we are thankful to have a chance to improve our manuscript and hope that it is good enough to be published in the Journal of Clinical Medicine

Please let us know if there are any more things we need to change.

Thank you again.

Point 1: Was the specificity of the antibodies used for IHC tested? For instance, did one specific band occur when performing western blot? This question came up when TCGA survival data of CRC patients were checked on the Human Protein Atlas website. Here, it seems like low SQLE expression is associated with poor prognosis. How do the authors explain this discrepancy?

Response 1: As a recommendation of the reviewer, we used IgG-rabbit polyclonal squalene epoxidase (SE) antibody (1:100 dilution; Interchim, Montluçon FRANCE). Before IHC, we have confirmed the molecular weight of SQLE (64kDa) by western blot. We rewrite information about primary antibodies.

Also, we checked the TCGA survival data of CRC on RNA level (http://www.oncolnc.org).  

High-SQLE shows worse prognosis than low-SQLE on m-RNA level; however, no significant correlation in clinical outcomes (Log-rank p-value = 0.896). 양식의 However, our IHC study shows SE is abundantly expressed at invasive front and high-SE is significantly correlated with some valuable clinicopathologic factors.

Because mRNA level does not usually predict protein level, there may be a difference in expression between the RNA and protein levels.

Our IHC study shows that patients with high-SEinvasive front have poor overall and disease-free survivals than patient with low-SEinvasive front (p < 0.001).

TCGA survival data of CRC on RNA level (http://www.oncolnc.org/kaplan/?lower=5&upper=95&cancer=COAD&gene_id=6713&raw=SQLE&species=mRNA)

Point 2: In general, many results were provided in the text (percentages, n-numbers) or in tables (e.g. Table 2, Table 3, section 3.2) but not in figures. However, it would be helpful to the readers if these results would also be visualized. For instance, the finding outcome of the assessment of the IHC could be shown in pie charts displaying: the IHC intensity (score 0-2) and the proportion of positively stained tumor cells (score 1-4) as described in section 2.2 (line 83-87). Another example would be Table 3 where the results could be visualized as a Venn diagram.

Response 2: As the reviewer recommended, we made a new Venn diagram of SQLE and E-cadherin expressions in the invasive front and tumor center (new Figure 2 and modified Table). Also, we exchange the order of Table 2 and Table 3 according to the order of the results.

While making the Venn diagram, we found an unintended error of the comparison of expression pattern between SQLE and E-cadherin in the SPSS statistics. We corrected Table 2 (old Table 3) and revise the manuscript.

Old Figure 2 will change to Figure 3 sequentially.

We mentioned the result of expression of SQLE in the tumor center, as follws:

“Also, of the 60 cases (42.0 %) with high expression of SE in the tumor center, 47 cases (73.8%) revealed the loss of immunoreactivity of E-cadherin.”

Table 2 (old Table 3). Comparison of expression between SQLE and E-cadherin in CRC

SE expression

E-cadherin expression

p-value

High (n=17) (%)

Low/negative (n=126)   (%)

Invasive   front

Low/negative (n=58)

3 (5.2)

55 (94.8)

<0.001

High (n=85)

14 (16.5)

71 (83.5)

Tumor   center

Low/negative, (n=83)

4 (4.8)

79 (95.2)

<0.001

High, (n=60)

13 (21.7)

47 (73.8)

SE, squalene epoxidase; p values were estimated by Fisher’s exact test and p <0.05 are highlighted in bold.

Figure 2. Immunohistochemical relationship between SE and E-cadherin in colorectal cancer. The expression pattern of SE is reversely correlated with E-cadherin expression in both invasive front and tumor center of colorectal cancer (p < 0.001).

Point 3: Can the authors provide survival data correlated to total SQLE staining intensity? In Figure 2 only survival data associated with high/low SQLE staining at the invasive front were shown. How was the survival affected when the staining in the complete tumor section (and not only the invasive front) was considered?

Response 3: As the reviewer pointed out, we can provide the clinical outcomes of SE expression at the tumor center as bellows.

Kaplan-Meier survival analysis by SE expression status in tumor center

High level of SQLE expression in the tumor center was significantly correlated with reduced overall survival (p = 0.018)(B). Also, high SQLE expression in the tumor center was correlated with poor disease-free survival, although it was no statistically significant correlation (p = 0.051) (A).

In this study, we could mention the survival rates of SQLE expression in the tumor center. However, our findings showed that high-SQLEinvasive front is associated with an EMT-associated marker and could serve as a possible therapeutic target in CRC patients.

To emphasize the better results, we focused the SQLE expression at the invasive front of the CRC.

Point 4: In Table 1 clinical features were associated to negative/low and high SQLE levels. Do these levels refer to the staining intensity at the invasive site of the tumor of at the total staining intensity?

Response 4: As the reviewer pointed out, SE level refers to immunostaining intensity at the invasive front. We add “invasive front” in all Tables

Point 5: It is mentioned that “Figure 1 and 2 show representative expression patterns of SQLE and E-cadherin in CRC.” (line 118-119), however, Figure 2 displays survival data.

Response 5: As the reviewer pointed out, we delete “and 2”.

Point 6: How long have the sections been autoclaved at 120°C when performing IHC?

Response 6: As the reviewer pointed out, slides were autoclaved at 120oC for 10 minutes. We modified the sentence of the manuscript “and autoclaved at 120oC for 10 minutes with 10 mM/L sodium citrate buffer (pH 6.0).”

Point 7: Please provide the registration number of the ethical permission.

Response 7: As the reviewer has mentioned, we have added the IRB number (IRB File No. 2019-02-014-001).

Point 8: The upper half of the Discussion (line 181-212) contains a lot of theoretical background on SQLE which (at least in some parts) should be moved to the Introduction section.

Response 8: As the reviewer recommended, we moved the theoretical background on SQLE to the Introduction section and modified in blue color

Point 9: Please add n-numbers to the Kaplan-Meier curves.

Response 9: As the reviewer pointed out, we added n-number in Kaplan-Meier curve on new Figure 3 A and B.

Point 10: The number of CRC patients analyzed in this study (n=143) should be mentioned in the Materials and Methods section.

Response 10: As the reviewer pointed out, we mentioned the 143 CRC patients in the Materials and Methods section.

Point 11: There is no section “2.1” in the Materials and Methods section. The first part of the Materials and Methods section could be subdivided into “2.1 Sample acquisition and tumor grading” and “2.2 Immunohistochemistry” which are then followed by “2.3 Assessment of immunohistochemical staining”.

Response 11: As the reviewer pointed out, we subdivided 2.1 to 2.5 as follows:

2.1 Case selection

2.2 Immunohistochemistry

2.3. Assessment of immunohistochemical staining

2.4. Statistical analysis of prognostic parameters

2.5. Ethical permission

Point 12: Add labels to Figure 1: On the left side of A) the label “SQLE” and on the left side of D) “E-cadherin” could be added. In addition, scale bars are missing.

Response 12: As the reviewer pointed out, we added labels of squalene epoxidase (SE) and E-cadherin, scale bars, and alphabets.

Point 13: Please revise the Acknowledgments section “Acknowledgments: In this section you can acknowledge any support given which is not covered by the author contribution or funding sections. This may include administrative and technical support, or donations in kind (e.g., materials used for experiments).” (line 252-254).

Response 13: As the reviewer recommended, we revised Acknowledgments: We appreciated Dr. Kim Nam Soon (Medical Genomics Research Center, Korea Research Institute of Bioscience and Biotechnology) for the intellectual discussions.

Point 14: Please define meanings of acronyms the first time they are being used.

Response 14: As the reviewer has mentioned, we checked acronyms and switched the wrong abbreviations.

Point 15: The authors did not pay attention to the nomenclature. Human gene names should be written in capital letters and be italicized. Human proteins are written in capital letters. Murine gene names should be written in small letters (only the first letter is capitalized) and be italicized. Murine proteins are written in capital letters.

Response 15: As the reviewer pointed out, we changed the nomenclature of wrong human genes and proteins.

Round  2

Reviewer 2 Report

All previous concerns and suggestions have been addressed sufficiently by the authors.